# Frequency and preventative interventions for non-suicidal self-injury and suicidal behaviour in primary school-age children: a scoping review protocol

Danai Bem,[1] Charlotte Connor,[2] Colin Palmer,[2] Sunita Channa,[2] Max Birchwood[2]

[1]Institute of Applied Health Research, College of Medical and Dental Sciences, University of Birmingham, Birmingham, West Midlands, UK
[2]Department of Mental Health and Wellbeing, Warwick Medical School, University of Warwick, Gibbet Hill Campus, Coventry, UK

**Correspondence to**
Dr Charlotte Connor; charlotte.connor@warwick.ac.uk

## ABSTRACT

**Introduction** Non-suicidal self-injury (NSSI) and suicidal behaviour have been witnessed in children as young as 6–7 years of age, but while there are many reviews of preventative interventions for NSSI and suicide in adolescents, few have explored its prevalence in younger children and the potential impact of preventative interventions at this stage of life. NSSI and suicidal behaviour are an increasing concern in schools but school-based programmes can improve knowledge, attitudes and help-seeking behaviours and help prevent escalation of NSSI and later suicide. This scoping review will aim to explore the nature and extent of the evidence on the magnitude of NSSI and suicidal behaviour in primary school children, and to examine whether there are any primary school-based interventions available for the prevention of this phenomenon in 5 to 11-year-olds.

**Methods and analysis** A scoping review will be conducted using established methodology by Arksey and O'Malley and the Joanna Briggs Institute. Multiple bibliographic and indexing databases and grey literature will be searched using a combination of text words and index terms relating to NSSI, suicide, primary schools, frequency and intervention. Two reviewers will independently screen eligible studies for study selection and extract relevant data from included studies. A narrative summary of evidence will be conducted for all included studies with results presented in tables and/or diagrams. Inductive content analysis will be used to understand any narrative findings within the included studies.

**Ethics and dissemination** Ethical approval is not required for this scoping review. The results of this review will be disseminated though publication in a peer-reviewed journal and presented at relevant conferences.

## INTRODUCTION

The National Self-Harm Network describes self-harm behaviour, or non-suicidal self-injury (NSSI), as 'the act of deliberately causing harm to oneself by causing a physical injury, by putting oneself in dangerous situations and/or self-neglect.'[1] Self-harming behaviours can include cutting or scratching the skin,

### Strengths and limitations of this study

► To the best of our knowledge there is no comprehensive review looking at the magnitude of non-suicidal self-injury and suicidal behaviour in 5 to 11-year-old children or any school-based interventions used for the prevention of these phenomena in this young age group.

► The proposed review will be based on established scoping review methodology, a comprehensive search, systematic screening carried out in duplicate and data abstraction employing both quantitative and qualitative approaches.

► As this is an unexplored topic relevant opinion articles, national statistics data and evidence from relevant charity and patient organisations will be considered for inclusion in an attempt to capture all available evidence in the literature.

burning/branding with cigarettes/lighters, scalding, overdose of tablets or other toxins, tying ligatures around the neck, punching oneself or other surfaces, banging limbs/head and hair pulling.[2] Common referred to as NSSI, it can also describe behaviours that may be culturally acceptable yet which lead to self-inflicted physical or psychological damage such as smoking, recreational drug use and excessive alcohol or body piercing, tattooing and body enhancement.[3] However, the latest edition of the DSM-5 (Diagnostic and Statistical Manual of Mental Health Disorders) does not include these culturally acceptable behaviours in their criteria.[4] NSSI and suicidal behaviour are often grouped together and although these 'acts' differ, specifically in terms of intent, NSSI has been shown to be a strong predictor of future suicide attempts in depressed young people.[5]

NSSI and suicidal behaviour are most commonly observed during

adolescence[6 7] with an estimated lifetime prevalence in high school students between 20% and 46%.[8] The average age of onset is around 12 years of age[9] and seems to be more common among women than men.[10 11] NSSI may indicate a temporary period of emotional pain or distress, or deeper mental health issues which may result in suicide. Both behaviours have been suggested to function as alternative coping mechanisms for adolescents who have poor emotional regulation strategies.[12–15] Indeed, NSSI has been witnessed in children as young as 7 years of age with 7.6% of third graders reporting self-harm engagement,[10] whereas 8.6% suicidal ideation was found among 6 to 9-year-olds with elevated aggressive-disruptive behaviour.[16] Being able to cope and manage emotions is something that increases with age, and NSSI and suicidal behaviour in young children may indeed represent a need to cope with difficult feelings with alternative methods in the absence of emotional regulation strategies.

With the majority of children spending much of their time in education, primary schools are well placed to identify younger children who may be at risk of developing NSSI and suicidal behaviour, and to implement interventions to support them in the development of strategies and skills to deal with life challenges and help prevent escalation into fully formed self-harm behaviour and potential crisis.[17] However, childhood self-harm and suicide remains a challenging topic for teachers who may not feel confident in dealing with such issues. Tools to support secondary school professionals and parents/carers are available and mainly focus on awareness and education programmes, screening, gatekeeper training, skills training and peer leadership.[18 19] Few studies, however, have explored the prevalence of NSSI and suicidal behaviour in primary school-age children[10 16] and the impact of preventative interventions at this stage of life.

This protocol outlines a scoping review that, unlike a systematic review, would not attempt to estimate the prevalence of NSSI and suicidal behaviour; instead it will systematically examine the evidence on the magnitude of those behaviours in younger aged children and how visible these behaviours are in the primary school setting based on educational professionals and school personnel's experiences. In addition, this review will aim to map and categorise the range of school-based interventions or programmes available to prevent NSSI and suicidal behaviour either by targeting the students themselves or by educating school professionals and parents to increase their knowledge and awareness on such phenomena.

### Research questions

This scoping review aims to identify the nature and extent of the evidence to address the following questions:

1. What evidence exists on the frequency of NSSI and suicidal behaviour in primary school-age children?
2. Are these behaviours witnessed by educational professionals and primary school personnel and to what extent?

3. What preventative interventions have been used in a primary school setting for NSSI and suicidal behaviour in younger aged children?

This protocol has been developed using the scoping review methodology proposed by Arksey and O'Malley[20] and further refined by the Joanna Briggs Institute (JBI).[21]

## METHODS AND ANALYSIS
### Search strategy
The following sources will be searched from inception up to the date the search will be conducted for identifying relevant studies:

► Bibliographic databases–MEDLINE, MEDLINE In-Process, EMBASE, PsycINFO, CINAHL, The Cochrane Library (CDSR, DARE), Applied Social Sciences Index and Abstracts, Education Resources Information Centre, British Education Index, Social Services Abstracts, and Social Policy and Practice
► Sciences and Social Sciences Citation Index (Web of Science) for citation searching
► The WHO International Clinical Trials Registry Platform, EPPI-Centre (TPoPHI) and ClinicalTrials.gov for ongoing studies
► Grey literature sources–Conference Proceedings Citation Index (Web of Science), Sociological Abstracts and relevant third sector organisation websites
► Checking of citation lists of included studies

The development of the search strategy will be a three-step process as described in the JBI guidance.[21] An initial limited search of MEDLINE and PsycINFO will be performed using keywords relating to the condition (NSSI and suicide) and the setting (primary school). Analysis of the identified terms in the title and abstracts of retrieved papers will be conducted by the primary researcher. A second-stage search will be performed across all included databases using a combination of indexing and free text terms found in phase 1 with search terms and strategies optimised for each database. For the first and second research questions relating to frequency and experiences of NSSI and suicidal behaviour, the search strategy will also combine terms relating to frequency/prevalence and experiences/believes; whereas for the third research question terms relating to the intervention (eg, universal, selective) will be used. Relevant keyword search terms are provided in table 1. Finally, reference list and third sector organisation website searching will be undertaken to identify any additional eligible studies. There will be no restriction on language or date of publication.

EndNote V.X7.4 (Thomson Reuters, New York) will be used to manage the records identified from the literature search and to record decisions during the study selection process. Two reviewers will independently screen studies for eligibility using predetermined inclusion/exclusion criteria. Any discrepancies between reviewers will be resolved with discussion or the input of a third reviewer

**Table 1** Keyword search terms

| Key concept | Keyword |
|---|---|
| Condition | (non-suicidal) self-injur*, (deliberate) self-harm*, self-mutilat*, self-neglect, self-cut*, parasuicid*, suicid* ideation/attempt/behaviour/think*/thought/plan* |
| School | primary, elementary, basic, grade, preparatory, first, middle, junior |
| Child | child*, kid, pupil, youth, youngster, junior, schoolboy, schoolgirl, schoolchild, juvenile, student |
| Frequency | frequency, prevalence, incidence, occurrence, trend, experience, believes, number, count, population at risk |
| Intervention | universal, selective, targeted, indicated, school-based, classroom-based, curriculum-based, psycho*, therap*, emotional, educational, gatekeeper training, skills training, social emotional learning |

*Truncation symbol/wildcard.

where needed. Translation of non-English language articles will be undertaken if necessary.

### Selection criteria

#### Types of studies

Any primary studies and reviews (systematic or narrative) on: (1) the frequency and school professionals' experiences of NSSI and suicidal behaviour in primary school-age children (quantitative or qualitative evidence) and (2) on primary school-based interventions for the prevention of NSSI and suicidal behaviour will be included (quantitative evidence). In an attempt to identify all sources of evidence for such an unexplored topic, opinion articles and relevant organisation surveys, national statistics data and reports will be considered for inclusion. Article authors will be contacted if there is a need for further information or clarifications.

#### Population

The review will consider all studies on the frequency of the phenomenon and the available interventions for its prevention that include children between 5 and 11 years of age within a primary school setting. This age range was selected taking into account the variety of primary education systems found in different countries. Studies on interventions that target teachers, principals or other school-related staff (eg, nurses, local community voluntary workers, governors) as well as parents, families and carers of 5 to 11-year-olds will also be included as long as they are delivered in the primary school setting.

The included studies could potentially focus on a range of vulnerable groups including: looked-after children; young carers; children who have experienced traumatic events (eg, emotional, physical or sexual abuse, bullying); learners with special educational needs/learning difficulties and disabilities; children with behaviour and attendance issues; children with mental health issues and chronic illnesses; children of parents with mental health issues; children of asylum seekers, refugees and new migrants; first nation/indigenous children; and Gypsy, Roma and Traveller pupils. Studies with mixed populations (eg, young children and adolescents) will be considered if they include results for the population of interest.

#### Concept

For the first research question all quantitative study designs reporting the frequency of NSSI and suicidal behaviour in 5 to 11-year-old children will be included. Measures of frequency will include (but not limited to) prevalence, incidence, trends and population at risk estimates.

Any qualitative study design that investigates the experiences of educational professionals (eg, educational psychologists, speech and language therapists, paediatricians) and primary school personnel (eg, teachers, nurses, local community voluntary workers) for NSSI and suicidal behaviour in primary school settings will be included.

For the third research question any primary school-based interventions for preventing NSSI and suicidal behaviour will be included. Educational settings with mixed populations of young children and adolescents (eg, middle schools) will be considered providing results for the population of interest can be extracted. Interventions included will be: (1) universal interventions targeted at all pupils and/or school staff, families, parents and carers (eg, promoting health and well-being, managing feelings, gatekeeper training, mental health awareness and skills training programmes); (2) selective interventions to prevent/reduce development of problems in pupils identified at risk; and (3) indicated interventions in order to prevent/reduce transition to serious health problems in pupils already presenting with low-level NSSI or show suicidal behaviours. Interventions may be delivered by school staff, other health professionals or third sector organisations in the primary school setting.

#### Context

Any primary or elementary education setting (eg, primary school, elementary school, lower primary school, first school, grade school) will be included. Three-tier educational settings where primary education starts at lower/first school (age range 5–9; depending on each country's system) and continues through the first grades of middle school (age range 9–13) will be considered on the bases that the reported data of interest are identifiable and extractable. Alternative educational settings (eg, Pupil Referral Units), special schools and homeschooling will be excluded.

## Charting the data

Charting will be conducted by one reviewer using a draft data charting form developed in line with the questions of the scoping review. Recorded information will be checked for accuracy by a second reviewer and any discrepancies will be resolved through discussion or referral to a third reviewer. The following data fields will be recorded from included studies: (1) general information including author, year of publication, country of origin, study design, setting, NSSI and suicidal behaviour definition; (2) population demographics for children sample (eg, age, gender, relation with any vulnerable groups); (3) population demographics for individuals describing experiences with NSSI and suicidal behaviour; (4) type of intervention, number of sessions, who delivered it, who is the targeted population; (5) data collection method, findings (numerical for quantitative studies or authors' themes and interpretations for qualitative studies), methods of data analysis. This list is indicative only as the charting process is iterative and the data extraction form may be further refined and updated at the review stage.[21] While the type of study design and methodology used will be listed, we are not excluding articles by study design, nor will we assess quality of evidence as the main aim of the scoping review is to have a broad knowledge of the available evidence and not present a view regarding the 'weight' of evidence (as recommended by Arksey and O'Malley).[20]

## Collating, summarising and reporting the results

A narrative summary of evidence will be conducted for all included studies. The study selection process will be illustrated using a flow diagram similar to the Preferred Reporting Items for Systematic Reviews and Meta-Analysis flow diagram including a numerical overview of the type of the included studies. Results from quantitative studies relating to the first (frequency) and third (interventions) review questions will be presented in tables and/or diagrams in order to map: the geographical distribution of studies; the range of study population (eg, age, socioeconomic status, vulnerable groups); measures of frequency used; the range of interventions delivered in primary schools and the measures of effectiveness used; the research methods adopted.

For the second review question (experiences) tables of descriptive characteristics of the included qualitative studies will be generated mapping: the geographical distribution of studies; the range of respondents; themes and subthemes reported by authors; the research methods adopted for data analysis. Inductive content analysis will be undertaken to analyse any narrative data in included qualitative studies and opinion articles. Common categories and themes within these categories will be identified and compared with those reported by the study authors. This approach to qualitative data analysis will allow the reviewer to gain information from the studies without applying predetermined codes or theoretical perspectives.[22] The results may also be grouped according to the type of the respondent (eg, educational professional, teaching staff).

Thus, the objective of this scoping review will not be to synthesise research findings in depth by employing meta-analysis or metasynthesis techniques but to visualise the range of evidence available in order to identify existing gaps in the literature, and reveal potential topics for future systematic reviews in the area of NSSI and suicidal behaviour during early childhood. It is hoped that this review will increase our knowledge of and help inform strategies to address this problem.

## CONCLUSION

This scoping review aims to examine the evidence that exists on the frequency of NSSI and suicidal behaviour in primary school children, and to explore whether there are any primary school-based interventions available for the prevention of NSSI and suicidal behaviour in younger aged children. Early childhood NSSI and suicidal behaviour have emerged as a critical issue in the last couple of decades but, as a phenomenon, are still relatively unexplored. This scoping review has the potential to identify gaps between level of need and relevant preventative measures adopted by schools in order to raise awareness, and may also contribute to the development of novel interventions for the prevention of NSSI and suicidal behaviour in this young age group.

**Contributors** CC and MB conceived the idea for the study. All authors collaboratively designed the study. DB led the development of the search strategy. CC and DB led the writing of the protocol. CP and SC critically reviewed the protocol. All authors approved the final version of this article.

**Funding** This study is funded by the National Institute for Health Research (NIHR) Collaboration for Leadership in Applied Health Research and Care (CLAHRC) West Midlands. The views expressed are those of the authors and not necessarily those of the NHS, the NIHR or the Department of Health.

**Competing interests** None declared.

**Patient consent** This is a scoping review protocol and primary data were not collected.

**Ethics approval** This scoping review does not require ethical approval as primary data will not be collected. This protocol could not be prospectively registered with PROSPERO as scoping reviews are currently excluded. The results of this review will be disseminated though publication in a peer-reviewed journal and presented at relevant national and international conferences.

**Provenance and peer review** Not commissioned; externally peer reviewed.

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
