## [Reviewer comments · BMJ Open]

ARTICLE DETAILS

TITLE (PROVISIONAL)	Frequency and preventative interventions for non-suicidal self-injury and suicidal behaviour in primary school-aged children: a scoping review protocol
AUTHORS	Bem, Danai; Connor, Charlotte; Palmer, Colin; Channa, Sunita; Birchwood, Max

VERSION 1 - REVIEW

REVIEWER	Sally Merry University of Auckland New Zealand
REVIEW RETURNED	27-May-2017

GENERAL COMMENTS	This is an important and under-researched topic and I will be very interested to read the results. The study is well explained. I have only one minor suggestion, and that is to add first nation people/indigenous people in your list of vulnerable people. In many countries including New Zealand, Australia and Canada, the rates of self-harm are very much higher in these groups.
---

REVIEWER	Alison Calexar Australian National University. Australia
REVIEW RETURNED	29-May-2017

GENERAL COMMENTS	The current manuscript presents the protocol for a scoping review on "frequency and preventative interventions for non-suicidal self-injury and suicidal behaviour in primary school-aged children". The protocol is comprehensive and clearly details the rationale for the scoping review and the proposed methods. The following minor changes are recommended: 1. Please specify the time period of the search (i.e., what will be the start and end date of the search?)2. In the keyword search terms, you could consider:- suicid* (to capture suicide, suicidal, suicidality)- suicidal thinking (in addition to suicidal ideation)- self-injury (in addition to non-suicidal self-injury)- child* (to capture child, children)- student (in addition to pupil)- number, count (in addition to frequency)- psycho* (to capture psychosocial, psychological)- therap* (to capture therapy, therapist, therapeutic)- "social emotional learning"
---

VERSION 1 – AUTHOR RESPONSE

Reviewer: 1

Reviewer Name: Sally Merry

Institution and Country: University of Auckland, New Zealand

This is an important and under-researched topic and I will be very interested to read the results. The study is well explained. I have only one minor suggestion, and that is to add first nation people/ indigenous people in your list of vulnerable people. In many countries including New Zealand, Australia and Canada, the rates of self-harm are very much higher in these groups.

We thank the reviewer for the positive comments and for the recognition that it is “an important and under-researched topic”. The reviewer’s suggestion for including indigenous people is very useful and we have amended our population inclusion criteria (see page 6) to include this vulnerable group.

Reviewer: 2

Reviewer Name: Alison Calear

Institution and Country: Australian National University, Australia

The current manuscript presents the protocol for a scoping review on "frequency and preventative interventions for non-suicidal self-injury and suicidal behaviour in primary school-aged children". The protocol is comprehensive and clearly details the rationale for the scoping review and the proposed methods.

We are pleased that the reviewer finds the protocol “comprehensive and clear”.

1. Please specify the time period of the search (i.e., what will be the start and end date of the search?)

We have mentioned in the search strategy section of the protocol (see page 5) that no restriction on date of publication will be applied. However, we have made this clearer by adding an additional statement (see page 4): “The following sources will be searched from inception up to the date the search will be conducted for identifying primary relevant studies”.

2. In the keyword search terms, you could consider:

- suicid* (to capture suicide, suicidal, suicidality)
- suicidal thinking (in addition to suicidal ideation)
- self-injury (in addition to non-suicidal self-injury)
- child* (to capture child, children)
- student (in addition to pupil)
- number, count (in addition to frequency)
- psycho* (to capture psychosocial, psychological)
- therap* (to capture therapy, therapist, therapeutic)
- "social emotional learning"

We have not provided a comprehensive search strategy with this protocol as this is going to be an iterative process with the key steps described in paragraph two of the search strategy section (see page 5). However, we have revised table 1 to incorporate the keywords suggested by the reviewer and some additional (i.e. suicid* thought/plan, population at risk).